# Comparative Economics of Conventional, Organic, and Alternative Agricultural Production Systems

**Timothy C. Durham [1],\* and Tamás Mizik [2]** 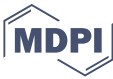

[1]  Agriculture and Biology Programs, School of Arts and Sciences, Ferrum College, Ferrum, VA 24088, USA
[2]  Department of Agribusiness, Corvinus University of Budapest, 1093 Budapest, Hungary; tamas.mizik@uni-corvinus.hu
\*   Correspondence: tdurham@ferrum.edu

**Abstract:** Agricultural production systems are a composite of philosophy, adoptability, and careful analysis of risks and rewards. The two dominant typologies include conventional and organics, while biotechnology (GM) and Integrated Pest Management (IPM) represent situational modifiers. We conducted a systematic review to weigh the economic merits—as well as intangibles through an economic lens—of each standalone system and system plus modifier, where applicable. Overall, 17,485 articles were found between ScienceDirect and Google Scholar, with 213 initially screened based on putative relevance. Of those, 82 were selected for an in-depth analysis, with 63 ultimately used. Economically, organic generally outperformed conventional systems. This is largely due to their lower production costs and higher market price. However, organic farms face lower yields, especially in the fruit, vegetable, and animal husbandry sectors. With that said, organic farming can provide significant local environmental benefits. Integrated pest management (IPM) is a potentiator of either core system. As a risk reduction and decision-making framework, it is labor intensive. However, this can be offset by input reductions without yield penalty compared to a conventional baseline. Biotechnology is a rapidly emerging production system, notably in developing countries. The use of GM crops results in lower production cost and higher yields. As a conventional modifier, its major advantage is scale-neutrality. Thus, smaller and lower income farmers may achieve higher gross margin. The main source of environmental benefits is reduced pesticide use, which implies a decreased need for fuel and labor. Barring external influences such as subsidies and participation in prescriptive labeling programs, farmers should focus on an a la carte approach (as opposed to discrete system adoption) to optimize their respective enterprises.

**Keywords:** agriculture; ecological; economics; conventional; organic; IPM; biotechnology; alternative; profitability; sustainability

## 1. Introduction

The continued sustainment of a rapidly growing population—in light of resource scarcity and environmental externalities—is perhaps the definitive challenge of the Anthropocene. This challenge can be partitioned into two components: food security and food safety. Food security is generally only a concern in developing countries, while developed and developing countries alike may face food safety issues.

Agriculture is the instrument to address said challenges. It enjoys unique standing as both a science and an art—with evident economic underpinnings. As a fusion discipline, its charge is deceptively simple: produce a larger quantity (security) and quality (safety) of food, fiber, and fuel to meet burgeoning demand. However, it's often argued that these goals are myopic, not only incompatible with environmental quality and human health, but irreconcilably so. Thus, agriculture is under frequent scrutiny. An introspective assessment necessitates a system that is profitable, sustainable, and attuned to diverse needs. As such, multiple competing systems have evolved. Each has a respective suite of strengths and weaknesses at the whole farm and aggregate level.

Conventional (often referred to as industrial) agriculture is the most common typology in developed countries. It is often seen as a natural outgrowth of Norman Borlaug's "Green Revolution". This system is large scale, dependent on inputs (synthetic fertilizers and agro-protectants) and highly mechanized. It has historically leveraged standard soil preparation methods (e.g., plowing followed by sowing). Yields per unit area are maximized. Despite evident merits, the counterargument is that food scarcity is illusory; wholly dependent on centralized food production/distribution predicated on the agro-industrial model. Indeed, questions abound about the long-term sustainability of such an approach.

Organics is a competing typology. Synthetic chemicals are generally prohibited, though naturally derived pesticides and fertilizers may still be utilized. Lower yields and greater labor needs are compensated for by higher prices. A frequent hallmark is diversification. This is leveraged as a risk mitigation tool, often entailing integrated crop/livestock production (as opposed to decoupled production in conventional spheres).

Organic food is often (mistakenly) considered to be "chemical-free", which is popular for certain consumer demographics. The focus on reconciling production with local ecologies also is attractive to consumers. Consumers are willing to pay a premium for these benefits, both real and perceived. Moreover, organic food (typically from developing countries) is often associated with fair trade labels that embed an ethic of social justice into the production system itself. It should be noted that there are multiple, organic inspired variants like biodynamics, regenerative, diversified, etc., though some of these systems may lack the heft of formal governmental standards and certification approval.

Integrated pest management (IPM) is a holistic system that arguably straddles the organic/conventional debate. Smith et al. (1976) provide an exhaustive overview of the origins. IPM is best described as a dynamic and situational portfolio of best management practices, supported by an economic decision-making framework. Rather than regular calendar sprays—irrespective of the presence/absence of a pest/pathogen—it's dependent on scouting and monitoring. This entails periodic, systematic assessments of crop status. Action is dictated by pest/disease thresholds. Moreover, grower tolerance for injury (quantity/quality impacts to a crop) and subsequent damage (actual economic impact borne by the farmer in the implementation/absence of action) factor into final management decisions. These strategically leverage a multifaceted management toolbox. This slate of options includes cultural, biological, and even chemical controls where warranted. The trade-off is that the system is knowledge intensive, which can complicate in-field actions and stifle adoption. Despite the potential for considerable cost savings, many farmers still prefer a predictable and reductionist calendar spray approach.

Biotechnology (often referred to as genetically engineered, genetically modified, or precision breeding) is a relatively new innovation. The first "modification" took place in 1944 at the Rockefeller Institute when a bacterium was transformed by extracted DNA (Brown and Fedoroff 2004). In 1994, a delayed-ripening tomato named the Flavr Savr received approval as the first GM product on the US market (Zhang et al. 2016). Biotechnology is a tool that accelerates breeding outcomes to develop crops with desirable agronomic traits. These traits are able to be "ported" between non-sexually compatible species, significantly broadening the cache of genetic material available to engineer new crop variants. Since the first widespread commercialization in 1996, the major aims of crop biotechnology have been increased pest and herbicide resistance, and higher tolerance (better adaptability). In the last two and a half decades, GM crop adoption has been rapid, especially in the Americas and South Asia. In 2019, 190.4 million hectares was planted in 29 countries, with higher growth in developing countries (Tome and Dionglay 2021).

The structure of this review is as follows. The next section introduces the article search and selection process. The third section outlines our results grouped into two subsections: a comparison of conventional and organic production systems and the economics of alternative agricultural production systems. With these systems identified, we outline findings from a systematic literature review on comparative economics. Where applicable, aforementioned variants or system "modifiers" such as biotechnology, IPM, and others

will be broached for context and inclusivity. The final section concludes and posits themes for further discussion.

## 2. Methodology

To appropriately consolidate the evidence and draw reliable conclusions (while keeping the sources manageable), an optimal n of 40 < n > 80 was established. Accessions for inclusion in our analysis were identified through keyword searches in the bibliographic database ScienceDirect. Post-search, "recommended articles" associated with each initial hit were also parsed for relevance, cross-checked against the original list of hits for duplication, and selected where appropriate. To ensure a breadth and depth of literature outside of the Elsevier publishing family, a recurrent search was also performed in Google Scholar, a more platform-agnostic option.

Prior to the execution of the search, primary and peer-reviewed credentials (including theses, dissertations, and Extension publications) were identified as a narrowing criterion for selection. An exception was made for peer-reviewed, secondary meta-analyses and book chapters that aggregated and reported on primary literature.

To holistically resolve the economic performance of comparative agricultural systems, search parameters included the terms conventional, organic, IPM, biotechnology, comparison, and economics. As core production philosophies, the selection of conventional and organic terms is self-explanatory. IPM and biotechnology were chosen as "bridging" terms. They represent a descriptive production modifier that can straddle (IPM can be practiced in both systems) or overlap and enhance a baseline philosophy (biotechnology is limited to conventional).

The Boolean operator "AND" separated each keyword query in the search field. Searches were not year or geo-limited, provided the study was in English. With that said, contemporary articles within the past 15 years were generally favored. Finally, only hits with the term "economic(s)" in the title were selected for further review. Overall, 17,485 articles were found between ScienceDirect and Google Scholar (not including recommended articles in ScienceDirect), with 213 (including recommended articles in ScienceDirect) initially screened based on putative relevance. Of those, 82 were selected for an in-depth analysis and 63 ultimately utilized.

Post-search results in ScienceDirect trended towards articles in academic journals, while Google Scholar included articles, book chapters, and so-called grey literature. The latter may encompass conference and working papers, as well as reports. Given that these are generally not subject to the same peer-review rigor, they were excluded from consideration.

In addition, subsequently culled articles generally focused on socioeconomic impacts to the consumer rather than the producer (outside the scope of this paper), found to be duplicative between ScienceDirect and Google Scholar post hoc, or failed to sufficiently broaden the system, commodity, and/or geographic representation herein.

Predictably, the final cohort skewed toward the agronomic and horticultural domains, though select livestock, dairy, and aquaculture articles also satisfied our stepwise screening process. These are outlined in Section 3. Additional references outside the confines of the original search have been added for context in Section 1.

## 3. Comparative Economics of Agricultural Production Systems

The analyzed articles were divided into 2 subgroups: comparison of conventional and organic production systems, and alternative agricultural production systems (including integrated pest management and biotechnology) along with discrete sections on ecological economics and socioeconomic measures contributing to the regulation of pesticides.

### 3.1. Comparison of Conventional and Organic Production Systems

3.1.1. General Comparisons

Organic farming is characterized by lower yields and input use, as well as higher output prices compared to conventional systems. This requires a greater reliance on natural

pest control methods. Cacek and Langner (1986) outlined that, despite reduced pesticide and fertilizer use, organic farms are less vulnerable to the same pest or unfavorable weather events due to diversification. This also provides a better seasonal distribution of both inputs and outputs, as well as more balanced income. However, organic farming requires better organizational skills from a management perspective. There are genuine environmental benefits of this production method at the farm level, but it's often hard to monetize and compare organic to conventional farming. Nguyen et al. (2008) underscored the importance of crop rotation in organic farming systems. Its advantages can be higher efficiency, lower $CO_2$ emissions, and long-term sustainability. Shorter food supply chains, such as farmers' markets, roadside stands, and community supported agriculture, help farmers to achieve higher income.

According to MacRae et al. (2007), organic yields, on average, are 10% less than conventional. This is higher in those countries where conventional farming is more intensive, e.g., in Europe, and for animal products (roughly 20%). At the profit level, the difference between organic and conventional systems can be ± 20%. Positive factors for organic production are lower operating costs, higher weather resilience, price premium, and shorter supply chain. In Europe, this list can be expanded by various financial supports for conversion and maintenance of organic status. Cavigelli et al. (2009) carried out a field trial on a study site of 16 hectares to assess the long-term economic performance of organic and conventional field crops. With an average price premium of 111–138% and lower production costs and yields, organic systems achieved 2.4 times greater net returns at lower risk.

Pimentel et al. (2005) compared organic animal-based, organic legume-based, and conventional systems for corn-soybean crop rotation. Under normal weather conditions, conventional crop yields were higher. However, organic systems performed much better (around 30%) under drought conditions. Overall, corn yield was only 3% lower on average during the analyzed 10-year period. Organic production can be characterized by lower total production cost, derived from higher seed and machinery cost, as well as lower fertilizer, pesticide, and labor cost. Although organic production typically requires more labor, peak periods differ from conventional. This in turn enables the hiring of a cheaper workforce. Taking into consideration the significant price premium (up to 140%), organic corn was 25% more profitable than conventional corn over a 10-year period.

### 3.1.2. Product Specific Comparisons

Qiao et al. (2016) compared organic and conventional tea farms in Wuyuan, China and Kandy, Sri Lanka. In both instances, organic production performed better economically. In Wuyuan this provided a small profit, while the conventional tea farm was unprofitable. In Kandy, both were profitable. However, organic households performed slightly better. They also noted that location plays an important role in organic farming. For example, Chinese organic tea farms are sited in high altitudes where pests are less common. This results in simplified and cheaper pest management. Another social advantage of organic farming is the higher employment relative to conventional systems.

Based on Bolwig et al. (2009), organic pesticides, mulching, animal manure, and composting were the most common practices of organic coffee smallholder contract farms in tropical Africa. Higher net coffee revenue (75% on average) motivated farmers to participate in an organic scheme. This impact was higher than the gains from the use of additional organic practices (low-cost and effective farming techniques). This was 9% per practice due to higher coffee yield. Contracting seems to reduce farmer uncertainty about net returns and provides a price premium if quality criteria are met.

Hernandez-Aguilera et al. (2019) analyzed smallholder coffee producers in three South American countries (Colombia, Mexico, and Peru). The common theme of these shade-grown systems was the use of birds as biocontrol agents against coffee borer pests. Results indicated that the potential price premium compensated for any loss attributed to

lower yields. However, the value of the avian-based, predatory ecosystem service should also be taken into account (see Section 3.2.3).

The Pacific Northwest (US) is a suitable testbed to examine the viability of organic blueberry production. DeVetter et al. (2015) noted that blueberry production in Washington State is expected to expand, due to robust demand, especially under organic management. Interestingly, these markets are often an outgrowth of existing organic farms seeking to further diversify. In 2014, nationwide retail fresh organic blueberry prices were 49% higher than conventional. Additionally, prices for organic freezer blueberries are often twofold (or more) those of conventional berries. Large retailers will often exclusively carry organic blueberries during certain windows if they can secure a predictable, high-quality supply. Organic yields are comparable to conventional yields. Variable costs and total costs in years 0 to 7 were 12% and 10% greater, respectively, under organic management. However, the complication is a dearth of efficacious and cost-effective pesticides to combat invasive pests. Approved herbicides are similarly lacking, with similar attributes. Growers are particularly sensitive to organic regulations that can exert an impact on production practices or costs (e.g., loss of sulfur burners to adjust irrigation water pH), and climate change is anticipated to introduce stressors that would prompt production adjustments (e.g., cooling systems and shade cover).

In a related study, Julian et al. (2012) found that cumulative net returns in blueberries after 3 years were negative, ranging from −USD 32,967 to −50,352/ha when grown on raised beds and from −USD 34,320 to −52,848/ha when grown on flat beds. This was influenced by cultivar, mulch, fertilizer rate, and source. Highest yields were obtained in plants fertilized with the low rate of fish emulsion or the high rate of feather meal, but fertilization with fish emulsion by hand cost (in materials and labor) as much as USD 5066/ha more than feather meal. The highest yielding treatment combinations (growing on raised beds mulched with compost + sawdust and fertilized with fish emulsion) improved cumulative net returns as much as USD 19,333/ha over 3 years.

Julian et al. (2011) conducted an establishment and production study to provide growers with a tool for economic management and decision-making. They noted that blueberries are expensive to produce, with profitability hinging on yield and price per pound. Yield for an established farm varies with cultivar/variety grown, soil type, and management practices. The number of years to reach full production is also highly heterogeneous, with some farms taking up to 7 years.

Weeds are consistently a limiting factor in organic systems. As such, the identification of effective weed control approaches is critical. Using tine cultivation combined with sweeps and hand weeding, Wann et al. (2011) assessed the effects on both weed control, productivity, and economics in peanuts. Net revenues for cultivated treatments ranged from USD 3333/ha to 3637/ha, exceeding the control (USD 1795/ha). Cultivation duration (4–5 weeks) improved peanut yield, grade, and net revenue, while frequency (1× to 2× a week) exhibited no significant effect. Thus, cultivation maximizes productivity and returns at current market premiums.

While the literature is generally robust with regard to crops and livestock, a dearth of information exists in the organic aquaculture arena. A meta-analysis by Gambelli et al. (2019) found a number of prevailing trends. Firstly, the profitability of organic aquaculture is not guaranteed for all species. Moreover, feed costs and low economies of scale might not be compensated for by price premiums. Additionally, a lack of homogenous organic standards is a major limitation, particularly for developing countries seeking export markets. With that said, organic aquaculture represents an opportunity for resource-strained farmers because it readily integrates into pre-existing farming practices.

### 3.1.3. Country Specific Comparisons

Jánský et al. (2003) compared costs and revenues of organic and conventional farming systems in the Czech Republic. The exhibited comparable costs, though higher revenues were realized for organic products. Although most pesticides and fertilizers cannot be used

in organic systems, services and other directs cost are higher. This research was carried out before the Czech accession to the EU; therefore, their financial data were influenced by the generous support system of the Common Agricultural Policy.

Brozova and Vanek (2013) analyzed the financial statements of 51 organic and 153 conventional farms in the Southern region of Czechia between 2008 and 2010. According to their results, the share of profitable farms was higher among the organic farms in all three years. However, it should be noted that without subsidies, the majority of both organic and conventional farms would incur a loss. This highlights the importance of EU subsidies. This impact is even greater (up to 20% of their income) for organic farms. Organic farms performed significantly better economically, measured in return on assets, return on equity, and even in total asset turnover.

Náglová and Vlasicova (2016) compared the economic circumstances of 273 conventional, 112 organic, and 4 biodynamic farms between 2007 and 2012. The majority of conventional farms produced cereals, while half of the organic and biodynamic farms had mixed production. According to the financial indicators calculated (return on assets, equities, and costs), organic farms performed the best, followed by biodynamic farms. Organic farms showed the greatest profitability growth rate in the analyzed period, e.g., +79% for return on assets. Although biodynamic farms are structurally related to organic farms, they performed worse financially even though they received the largest amount of subsidy per hectare. Conventional farms had the lowest return on assets and equities, and the highest asset turnover. Contrary to the generally accepted opinion in the literature, organic farms had much lower labor costs than that of the conventional farms. The authors attributed this to the high share of family (unpaid) labor, while conventional farms use mostly paid labor. Similarly, lands are generally owned by organic farms, while larger, conventional farms typically rent. Biodynamic farms had the lowest total costs, as well as the lowest operating revenue and profit.

Krause and Machek (2018) compared 291 organic and 4045 conventional farmers in 2009–2013 to study financial differences. As organic farming relies more on mechanical rather than chemical protection, their input costs are lower, but their labor cost is higher than those of the conventional system. They found that Czech organic farms could reach only 41% of conventional cereal yield and 39% of the conventional potato yield. These values are extremely high compared to the literature, e.g., MacRae et al. (2007). According to their results, organic farmers, on average, had higher profitability, leverage, and firm size, but lower asset turnover. The latter contradicts Brozova and Vanek (2013), despite the fact that year 2009 and 2010 were part of both analyses. This can be explained by their different sample and geographical coverage. Based on a regression analysis, organic farms proved to be more profitable with significantly lower profit margin volatility. However, the vast majority of these results are in agreement with the results of Náglová and Vlasicova (2016); the only exception being the lower debt ratio of the organic farms.

Acs et al. (2007) calculated optimal resource use in the Netherlands. According to the authors, organic farms' labor and organic matter input was higher, while manure purchase was lower compared to conventional farms. Due to the strict rules of organic production, artificial fertilizers and pesticides were not purchased. Fixed costs of organic production were slightly higher than that of conventional systems, while variable cost was much higher due a very high labor cost. However, the high price premium was not only able to offset these higher costs, but also rendered organic production more profitable relative to conventional production. Based on their model calculations, this resulted in 2.5–2.7 times higher family labor income per hectare. However, there are considerable uncertainties related to organic production, such as the significant need for hired labor, as well as yield and market price risks. The conversion period amplifies these further. During this window, further financial problems may arise.

Binta and Barbier (2015) collected primary data from 20 organic and 20 conventional vegetable producer farms in Senegal. They did not find significant difference in labor level. The potential reason is the nature of production, because vegetables are inherently labor

intensive, even in conventional systems. Yield difference was small for onion (7%) but extremely high in tomato (64%). Although organic production costs were smaller, there was no price premium for them. This reduced gross margin for every vegetable crop evaluated. Further, there is only a limited market; consumer awareness of organic products is low in Senegal, which makes market entry difficult.

Aslam et al. (2020) analyzed 153 organic and 147 conventional wheat farms in Pakistan. At input level, irrigation and labor costs were higher at organic farms, while other expenses (fertilizer, and especially pesticide costs) were lower. Overall, total production cost of organic farms was 14–40% lower compared to conventional farms, with 14–23% lower wheat yield.

Pimentel (1993) compared US organic and conventional farming. He noted higher labor and lower fertilizer and pesticide use for organic farms. Surprisingly, organic maize yield was higher compared to conventional; mainly due to more effective pest control (crop rotations vs. the use of insecticides). This resulted in lower production cost (0.05 USD/kg vs. 0.08 USD/kg). However, this was not the case for vegetables and fruits, where the yield difference can be extremely high (e.g., 100% for potato). In addition, crop rotation itself is not wholly effective against insects and diseases.

Using Altman's model and Index IN95, both organic and conventional typologies in winemaking were assessed as "thriving" by Vlašicová and Náglová (2015), though the Ch and G indexes classified them in the grey zone. Organic enterprises were identified as exhibiting modestly better fundamentals. Profitability and solvency were higher, with lower indebtedness. It should be noted that their profit and subsidies per hectare were almost twice that of conventional, so corrections for this would provide more definitive comparisons. Despite this, both organic and conventional winemaking enterprises were capable of sound financial management and returns without the aid of any subsidies.

Sgroi et al. (2015) compared organic and conventional citrus systems on the Sicilian coast. Though the scale of the aforementioned study was minimal (one hectare area assessed), a higher economic and financial sustainability was noted for organics. The higher profitability of organic farming was due to the minor labor requirement and higher price premiums afforded in the market. It was suggested that organics could contribute to a revival of citrus farms in Sicily, with generational turnover providing the requisite infusion of entrepreneurial initiative.

Tanrivermiş (2008) compared organic and conventional hazelnut production from questionnaires and personal interviews in the Black Sea region of Turkey. The results of a three-year study were mixed and region dependent. Organic yields were 12.4% higher in region 1, and 1.2% lower in region 2, compared to conventional farms. The price garnered by organic farmers was 8.1% and 1.5% higher, respectively, than that received by conventional producers. The gross margin per hectare of plantation area in organic farming was 12.0%, and the net margin was 117.7% higher than that of conventional farms in region 1, whereas they were 0.3% and 2.2% higher in region 2.

Deka and Goswami (2021) emphasized that small-scale tea growers often resort to unsustainable agricultural practices. However, some growers are converting to sustainable production systems, including organics. Using primary data of small-scale tea growers in Assam India, a collective case study approach and mixed methods were used to evaluate the economic sustainability of organic tea cultivation on a small scale. Yearly income from a hectare of organic tea cultivation was valued at USD 1156.83, surpassing that of conventional (USD 1047.95) under one assumption; that organic growers could match the average conventional yield of 15,000 kg. However, given a real-world yield penalty of 10%, this organic value was reassessed at USD 973.10. Despite that, organic cultivation was found to be an economically viable long-term option for small-scale tea growers over a period of 10 years; provided the yield stabilized post-organic conversion. Additional income could be derived through optimal resource use, best farm management practices, and an organic premium on green leaves.

Though more limited in scope, comparison between organic and conventional livestock/dairy operations also denote trends. The improvement of functional traits through directed crosses and heterosis is a frequent focus of dairy breeders. The benefits span both conventional and organic approaches. Clasen et al. (2019) simulated the impacts of crossbreeding between Swedish Holstein and Swedish Red on herd dynamics and profitability in Sweden. An examination of both conventional and organic herds found that crossbreeding increased the annual contribution margin per cow by EUR 20 to 59. Increased profitability was heavily influenced by improved fertility. Replacement rate in the conventional systems was 39.3% in the pure-breeding strategy, decreasing to 35.8 and 30.1% in the terminal and rotational crossbreeding strategies, respectively. Similar trends were observed in the organic production system. Overall, crossbreeding strategies earned an additional EUR 22 to 42 per annum per cow from selling live calves for slaughter due to the extended use of beef semen. Milk production was similar between pure-breeding and terminal crossbreeding, and marginally decreased 1 to 2% in rotational crossbreeding.

One measure of sustainability is land utilization. While ill-suited for cropping, mountainous and semi-mountainous areas are ideal for grazing livestock. Organic sheep farming represents a promising alternative to conventional approaches in Greece. Tzouramani et al. (2011) analyzed the financial performance of sheep breeding and the risks assumed by producers. Using a stochastic efficiency analysis with respect to a function, economic viability of conventional and organic sheep farming was explored, as well as key factors determining economic outcomes. While both farming systems were viable, the viability of organics was mainly embodied in subsidies, while conventional farming generated a lower net return in the absence of payments.

### 3.1.4. Transition from Conventional to Organic Farming

A transition from conventional to organic farming is not limited to a single action; there are cascading effects. As seen previously, this requires a change in attitude, and a departure from (now) prohibited methods and inputs (synthetic chemicals) to a more labor-intensive model. Transition can take 1–3 years, depending on the product and government regulations. During this period, organic production methods must be observed. However, transitory products cannot be sold as organic. Practically, this means lower yields without a price premium.

Qiao et al. (2016) highlighted that a switch to organic production may be more advantageous for small-scale farmers in developing countries. They can attain higher yields through better seeds, organic fertilizer, and technical assistance. Although organic farming is labor intensive, lower input cost can offset it. Nevertheless, there are inherent risks in organic conversions. Although Acs et al. (2009) noted that organic yields do not necessarily fluctuate more than conventional yields, price risks are higher due to the small-scale and immature nature of organic markets.

Forster et al. (2013) examined agronomic and economic data from the conversion phase (2007–2010) of a farming systems comparison trial in central India. A cotton–soybean–wheat crop rotation under biodynamic, organic, and conventional (with and without Bt cotton) management was investigated. A significant yield gap between organic and conventional farming systems was evidenced in the 1st crop cycle for cotton (229%) and wheat (227%). However, that differential narrowed appreciably in the subsequent crop cycle. On average, conventional farming systems achieved significantly higher gross margins in cycle 1 (+29%), whereas in cycle two gross margins in organic farming systems were significantly higher (+25%) due to lower variable production costs (but similar yields). The authors noted that organic farming systems were less capital intensive than conventional ones. This may be of particular interest to smallholder farmers who typically do not have the financial means to purchase inputs and would otherwise need to seek loans. Thus, organic farmers might be less exposed to financial risks associated with fluctuating market prices of synthetic fertilizers and agroprotectants.

Jat et al. (2014) witnessed a phased inflection of yield and profitability. Though not a conversion per se, they examined seven permutations of tillage, crop establishment, and residue management in a rice–wheat rotation among poorly resourced farmers in South Asia. Though it initially lagged behind the conventional (CT) treatment, the yield and economic advantages of the conservation agriculture (CA) treatment was realized after 2–3 years.

Amid market overproduction and shuttering dairy enterprises, the price premiums associated with organic milk production represent an attractive opportunity for dairy-ers. Butler (2002) sought to identify the main determinants of production costs between conventional and organic systems. Specifically, whether differences were attributed to government-mandated requirements, optional procedures, or personal preferences. The investigation showed that the total cost of production on a per cow and a per hundredweight basis is 10–20% higher for organic producers. Accounting for those differences were higher feed costs, higher (average) labor costs, significantly higher herd replacement costs, and significant transition costs. Previous studies had indicated the higher net returns observed on organic enterprises are due mainly to a lower cost of production, which was not borne out by the results of this study.

Using propensity score matching, Mayen et al. (2010) derived a value of 13% reduced productivity when comparing organic to conventional dairy systems. However, they found little difference in technical efficiency between the respective systems when measured against appropriate technology.

Farm structure also dictates viability in the dairy sector. Mayen et al. (2009) estimated a multi-stage, multi-output cost function in order to measure vertical economies of scope in organic and conventional dairy farms. The model focused on the integration of on-farm grain and forage. While there were minimal vertical economies of scope for conventional dairies, significant ones were noted for organics. These were consistent with higher costs of obtaining organic feed through open market transactions (which is chronically underdeveloped). However, depending on scale, this degree of vertical integration may violate many of the social precepts of organics. Gillespie and Nehring (2012) compared the economic performance measures of organic and transitioning cow calf operations to conventional farms. A method of matching samples was used for the comparison to better assess comparative differences. In particular, each organic farm was matched with conventional one occupying a similar industry segment, farm size class, and region. Farmer demographic, system, and technology variables were further used to identify matches. The results suggested that the higher cost of organic production is attributed primarily to higher capital recovery, taxes and insurance, and overhead costs. Moreover, higher returns in organic enterprises were not noted.

With a focus on policy instruments that could be used to facilitate organic conversion, Demiryurek and Ceyhan (2008) compared organic and conventional hazelnut producers from a tripartite perspective: socio-economic characteristics, production systems, and economic performance. Survey results revealed that organic hazelnut producers were more educated, had larger hazelnut areas in production, and spent more time on agricultural activities. Cluster analysis was used to define comparable farmers from both production systems to compare variables. Organic producers required more labor, and used more lime, organic fertilizer, and insect traps, while conventional producers relied on synthetic inputs. Organic producers had lower costs of production and higher income.

### 3.1.5. Fair Trade Certification

Often heralded as economics with an environmental and social conscience, certified fair trade labels seek to embed an ethic of justice in support of marginalized producers. In addition to assurances that environmental and occupational health standards are met, a fair wage is guaranteed to workers/producers. Whether or not this makes "economic sense" while satisfying an oft nebulous sustainability benchmark is hotly contested in academic and policy circles.

Fair trade certification may provide further benefits during the transitional period of organic conversion, through cheaper group certification, and the even higher price premium. However, farm size may limit these benefits (Qiao et al. 2016). Using survey data from coffee growers in Mexico and Peru, Barham and Weber (2012) examined the economic sustainability of Fair Trade/organic and Rainforest Alliance schemes, while comparing them to a conventional standard. Interestingly, the analysis indicated that yields, rather than price premiums, were the primary driver for increasing net cash returns to these households. As such, it was suggested that certification norms that provide latitude for improved yields be stressed to enhance grower welfare and attract/maintain new participants.

While providing historical context for the growth of fair trade, Dragusanu et al. (2014) examined empirical evidence, based primarily on conditional correlations. The assessment suggested that Fair Trade does achieve many of its stated goals. However, this is on a relatively modest scale relative to macro-scale national economies. In general, Fair Trade participants command higher prices, improved credit access, deem their economic situation as more stable, and tend to engage in more eco-friendly farming practices.

Despite this, there are some lingering uncertainties. Farmers in Fair Trade cooperatives may not be fully cognizant of the workings of the business, leading to targeted mistrust for those who run the cooperative. There are also trade-offs between limiting certification to small-scale, disadvantaged producers and by allowing plantation-style producers to participate. Such scale-ups may erode some of the initial benefits of participation in the first place. According to the authors, the largest potential benefit of market-based systems is that they do not distort incentives like foreign aid does. Instead, Fair Trade works within the boundaries of the marketplace, rewarding productive activities, processes, and ethics that consumers value—the embodiment of voting with one's wallet.

### 3.2. Economics of Alternative Agricultural Production Systems

3.2.1. Integrated Pest Management

A review by Pimentel et al. (1992) suggests that it is technologically feasible to reduce pesticide use in the United States 35–50% without reducing crop yields. Indeed, the reduction of pesticide use is often an implicit goal of IPM systems.

Due to the constraints of pest monitoring, insecticide applications are generally conducted on a calendar schedule. Stephenson et al. (2019) assessed threshold-based management strategies, including the use of conventional-threshold and organic-threshold pesticide use relative to a calendar-based approach in tomato. Yield, management cost, and production value were quantified. The greatest total and marketable yields were obtained via conventional pesticides using action thresholds. This endorsed an IPM strategy in small-scale vegetable operations. A threshold plus organic threshold approach did not exert an effect on yields compared to a calendar-based approach. Fruits deemed unmarketable were greater with the use of organic insecticides; attributed to reduced efficacy and control residual. Production costs for the organic-threshold approach were also greater due to increased number of insecticide applications required. Gross margin for both conventional and organic threshold-based approaches were greater than for the conventional calendar method. Increased economic returns for conventional threshold was due to increased yields. An increase in return for organic threshold management was based on premiums received for organically grown tomatoes.

Integrating perennial crops into organic farming systems can be profitable, improve soil quality, and supply nitrogen to succeeding grain crops. Wachter et al. (2019) conducted a 5-year study examining four contrasting farming systems in dryland eastern Washington State. The systems included a conventional (CONV) winter wheat/spring wheat/spring pea rotation; a mixed crop-livestock (MIX) winter wheat/spring wheat/grazed winter pea forage rotation; an organic mixed crop-livestock (ORGcrop) rotation of 3 year perennial alfalfa and grass/grazed pea forage/winter wheat; and an organic hay (ORGhay) continuous perennial alfalfa and grass system. Over the 5-yr rotation, average net returns were

ORGhay (USD 616 yr) > ORGcrop (USD 216 yr) > MIX (−1 yr) = CONV (−13 yr). It should be noted that this was attributed, at least in part, to comparatively high hay prices and average grain prices compared to long-term averages. A comparable delineation series was also noted when examining soil sustainability metrics.

The identification and modality of pest management strategies are dependent on pest pressure. Crop rotation and diversification are the most commonly used tools for pest and disease control. However, large-scale agriculture takes advantages of specialization, rather than diversification. Bowman and Zilberman (2013) provided a perspective on diversified farming systems (DFS) that lie somewhere between organic and industrial agriculture. In general, multiple crop systems face many disadvantages, e.g., higher cost of tilling and harvesting, or crop insurance systems that discourage diversification. DFS also lacks consumer recognition.

When weighing alternative production methods, management and profitability decisions are paramount. Knudsen (2015) found that consumers in Northern Utah were willing to pay a premium for peaches grown using organic and "eco-friendly" production practices over conventional systems. Of the three aforementioned systems, organic had the highest average grower net returns and had the lowest associated risk, while conventional peach production had the potential for the highest net returns.

Contemporary literature increasingly strives to include return on investment (ROI) data beyond one-dimensional economic analyses. For example, xenobiotics in the environment attributed to chemical management of pests and diseases.

Rysin et al. (2015) investigated the economic viability and environmental impact of three different soil management systems used for strawberries in the southeastern United States: a conventional production system modeled on current production practices, a nonfumigated compost system with summer cover crop rotations and beneficial soil inoculants, and an organic production system adhering to practices approved under the National Organic Program (NOP). After developing enterprise budgets under a series of assumptions, all three systems resulted in positive net returns estimated at USD 14,979, 11,100, and 19,394 per acre, respectively. Moreover, using selected indicators, the nonfumigated compost system and organic system also resulted in considerable reductions in negative environmental and human health impacts. For example, the total number of lethal doses (LD50) applied per acre (summation of all agroprotectants) associated with acute human risk. This value declined from 118,000 doses/acre in the conventional system to 6649 doses/acre in the compost system and 0 doses/acre in the organic system.

IPM has oriented farmers to more environmentally friendly practices. In particular, it is a useful tool for producers in transition to organic farming. However, the extent of its economic impact is often poorly understood, thus hindering adoption. Scouting is a foundational element of IPM that provides baseline information to formulate an action plan. The goal is to sample a percentage of plants (relative sampling) that's representative of the pest pressures of the whole (field), without having to allocate scarce resources to conduct unreasonable whole field, plant by plant sampling (absolute sampling).

Ferrer and Hammig (2013) took this efficiency an intuitive step further. They examined the savings and potential profitability of an alternative scouting method, the binomial sequential scouting method (SSM), to conventional sampling (CS) in collards. SSM is a recent innovation for traditionally operated collard farms. It is geared toward a more economical execution without sacrificing procedural effectiveness. Analysis indicated that both scouting methods would result in cost savings on traditionally operated farms. In particular, the cost savings per hectare generated from IPM with SSM [3.62% of total cost (TC) and 3.91% of total variable cost (TVC)] is higher than the cost savings from IPM with CS (2.91% of TC and 3.15% of TVC). The difference in cost savings between CS and SSM was attributed to the expedited time window associated with SSM, thus lowering lower labor costs. Despite the seemingly minor cost savings, this is an appreciate multiplier at the farm and aggregate/state level.

The nexus of conventional systems with IPM and biotechnology modifiers also deserve careful consideration. Onstad et al. (2014) conducted a profitability evaluation of insecticidal Bt corn expressing cry (crystal) protein insecticide (a PIP, plant incorporated protectant, essentially a biodegradable insecticide), along with a "refuge".

As insects feed on corn with the insecticidal protein, their susceptibility generationally shifts (through artificial selection) to resistance over time, negating the long-term utility of the approach. As such, the refuge is a mitigation measure, a deliberate staging area to cultivate a susceptible population. These individuals spread throughout the field and mate with resistant populations, diluting resistance frequencies in the overall population and prolonging the lifespan of GM technology.

Using a published biological model and economic algorithm, the researchers evaluated refuge sizes of 5–50% of field area for single-trait Bt corn and 5–20% for pyramided Bt corn with two traits (each with a subtly different mode of insecticidal action) targeting western corn rootworm. They also considered the role of block and blended (a GM and non-GM seed mixture) refuges for insect resistance management (IRM). Results demonstrated that, for pyramided Bt corn, block refuges planted in the same location within a field year after year yielded the greatest overall profit. If growers relocated their block refuge annually, then a 5% blended refuge gave the greatest return. For single-trait Bt corn, 10–20% blended refuges gave greater economic return compared to block refuges ranging from 5% to 50%. Single-trait Bt corn with 5–20% block refuge (with no insecticide) was superior to soil insecticide use alone in all cornfields. Thus, while farmers are essentially using a continuous "calendar spray" with an endogenous plant-based insecticide—in abeyance with the standard conventions of IPM—to manage a soilborne pest, the strategies to mitigate resistance were effective. They also avoided the inherent labor costs and knowledge intensive attributes of scouting. Moreover, externalities associated with traditional soil applied insecticides were also evaded.

Reitz et al. (1999) developed and implemented a pesticide input management system for celery. The overall effectiveness was compared with a conventional pesticide application program and an untreated control for over 4 years in field station trials, and then implemented in a commercial trial. The low-input program relied on biological control agents and rotations of selective, environmentally safe biorational insecticides (*Bacillus thuringiensis*, spinosad, and tebufenozide) applied only when pests exceeded threshold levels. The conventional program included prophylactic applications of broad-spectrum synthetic pesticides. Yield losses from key insect pests were documented, with economic analyses comparing monetary returns derived from each program. Overall insect damage was lower for conventional program in only one of the four years. The IPM program utilized significantly fewer applications, but there were no significant differences in the total number of marketable cartons. These lower insecticide costs translated to greater net profits for the IPM program.

Though IPM is frequently associated with pests (as per its namesake), it also enjoys broad applicability to diseases caused by plant pathogenic microorganisms. A parallel commercial trial by Reitz et al. (1999) included a low input program for managing the fungal pathogen *Septoria apiicola*. The IPM program used over 25% fewer pesticides than the grower's standard program, and pest management costs were over USD 250/ha lower for the IPM program.

Integrated crop management (ICM) is often considered synonymous with IPM. More appropriately, it can be contended that it represents an extension of the best management practices concept with a more holistic lens, including soil properties and nutrition. Wani et al. (2017) conducted a study to understand soil properties, crop yield, and economics as affected by ICM. Balanced fertilizer application, both in rainfed and irrigated areas, directly influenced crop yields. Yields of cereals, legumes, and oilseeds were 3590, 1400, and 2230 kg/ha with improved management practices, compared to 2650, 1030, and 1650 kg/ha with conventional farming practices, respectively. Moreover, average net income estimated from conventional farming was Rs. 26,290/ha, while it was Rs. 35,540/ha

from improved management practices. This indicated that ICM practices resulted in 35% greater income. Oilseeds achieved higher net income and benefit–cost ratio, while the cereals and legumes also showed significant improvement in yield. The detailed findings on soil properties, yields of crops, and economics suggested that there is a vast potential for crop productivity improvement through ICM practices across different soil types and rainfall zones.

Assessment metrics extend well beyond the realm of economics. In intensive vegetable production, low organic matter (OM) inputs and nitrate (NO3-N) leaching degrade soil quality over time. Jackson et al. (2004) compared four management regimes for their effects on soils and production: a minimum tillage with OM (+OM) inputs; minimum tillage with no OM (−OM) inputs; conventional tillage +OM inputs; and conventional tillage −OM inputs. The addition of cover crops and compost increased microbial biomass C (MBC) and N (MBN), reduced bulk density, and decreased the NO3-N pools in the root profile. Thus, leaching potential was lower compared to −OM treatments. Minimum tillage tended to decrease lettuce and broccoli yields, but was not associated with increased pest problems. Weed density of shepherd's purse and burning nettle were occasionally lower in the +OM treatments. Disease and pest severity on lettuce was slight in all treatments, but for one date, corky root disease was lower in the +OM treatments. The Pea Leafminer was unaffected by management treatments. Economic analysis of the three lettuce crops showed that net financial returns were highest with minimum tillage −OM inputs, despite lower yields. These tradeoffs suggest that farmers should alternate between conventional and minimum tillage, while frequently adding OM, to enhance several indices of soil quality, and reduce disease and yield problems inherent with continuous minimum tillage.

### 3.2.2. Biotechnology

van den Bergh and Holley (2002) systematically collected arguments in favor of and against biotechnology. Pros were classified into three groups: environmental (e), human–economic (h–e), and food security in the developing countries (f). Argumentations included higher tolerance (e), improved food quality (h–e), and more stable/efficient agriculture (f). They grouped cons into four categories: ecological and environmental impacts (e–e), human health impacts (h), population growth (p), and social–economic (s–e). The major consequences highlighted were unwanted horizontal gene transfers and resistance (e–e), potential allergic reactions and antibiotic resistance (h), higher need for accessibility, distribution and sustainable production (p), and dependency on large, multinational companies, as well seed importation (s–e). They claimed that the advantages of genetic modification overstated and highlighted the increasing difficulties of consumers' free choice due to the large-scale cultivation of GMOs.

Flannery et al. (2004) carried out an economic cost–benefit analysis on five hypothetical GM crops: two traits of winter wheat, spring barley, sugar beet, and potato. Compared to their conventional production, GM crops would provide lower production cost and higher gross margin. Sugar beet performed the best with 6.06% lower production cost and 9.69% higher production margin. The latter would increase to 25.29% when yield increase was also taken into account. Theoretically, production of all analyzed crops would be beneficial at farm level. The relative size depends on seed and coexistence costs and the costs of pesticide use and/or higher yields.

Brookes and Barfoot (2018) provided an overview of economic and environmental impacts of GM crops from 1996 to 2016. According to their calculations, it resulted in an 18.2 billion USD increase of the global farm income in 2016 owing to higher productivity (up to 15% higher yields on average) and efficiency. More than half of this amount was realized in developing countries. Environmental gains were associated with lower pesticide use that requires less fuel due to fewer spray passes, and also facilitates the use of "no-till" and "reduced-till" farming practices. At country level, the major beneficiaries were the USA (79.5 million USD), Argentina (23.7 million USD), India (21.1 million USD), Brazil (19.8 million USD), and China (19.6 million USD) in the period of 1996–2016. At production

level, positive yield impact was the highest for maize (404.9 million tonnes), followed by soybean (213.5 million tonnes), and cotton (27.5 million tonnes) in 1996–2016.

Smale et al. (2008) examined the literature on the economic impacts of GM crops in developing countries. They noted that farm level impacts are assessed most frequently. Although farmers' profit was positive on average, some were disadvantaged by planting the more expensive seed. Even in the case of genetically engineered crops, there is a need for proper knowledge-based integrated pest management. Besides economics, other impacts should also be analyzed, such as on labor, health, and environment. Catacora-Vargas et al. (2018) analyzed the socio-economic impacts (SEI) of genetically engineered crops with a systematic literature review. They concluded that most of the analyzed research lack social and non-monetary economic aspects, and often generalize short-term and small-scale results.

Gómez-Barbero and Rodríguez-Cerezo (2006) noted that the use of GM crops may contribute to crop management simplification. It could also be a primary driver of higher off-farm income due to increased discretionary time. They also highlighted cost savings (lower weed (HT) and pest (Bt) control costs), as well as yield increases for certain crops. These impacts can greatly vary geographically. GM adoption rates are independent of farm size. Moreover, smaller and lower income farmers may achieve higher gross margin. Regarding benefit distribution, farmers are generally the main beneficiaries, followed by seed producers, and finally consumers due to lower market prices. Qaim (2009) argued that the major advantage of GM crop technology is scale neutrality. With this said, more attention should be paid to other elements, e.g., socio-economic, environmental, and health impacts. He further summarized the double impacts (insecticide reduction and increase in effective yield) for Bt cotton and Bt maize, finding an increase in gross margin up to 470 USD/ha; significantly positive in every analyzed country and product specific case.

Coexistence and labeling are two prominent issues regarding GM crops. The former deals production related issues, i.e., how to produce crops under different farming systems while avoiding potential contamination. The latter provides information to the consumers. This is particularly important in those countries that operate process-based regulation systems, e.g., the European Union, where the maximum share of GM components without labeling is 0.9% (adventitious presence). Greene et al. (2016) emphasized that the coexistence of organic, conventional, and GM products is challenging due to accidental pollination during production and mix-ups later in the supply chain. This increases the costs of organics due to strict labelling requirements that preclude "GM contamination" (see adventitious presence). The most commonly used practices against pollination are buffer strips and delayed planting. Despite these preventative measures, 1% of all U.S. certified organic farmers experienced this problem between 2011–2014. As a matter of labeling, McCluskey et al. (2018) argued against mandatory non-GMO labeling as that would increase their prices and, therefore, would favor GMO products.

Figure 1 gives a graphical representation of the analyzed competing production systems based on their major characteristics. Overlapping domains represent cross-functionality.

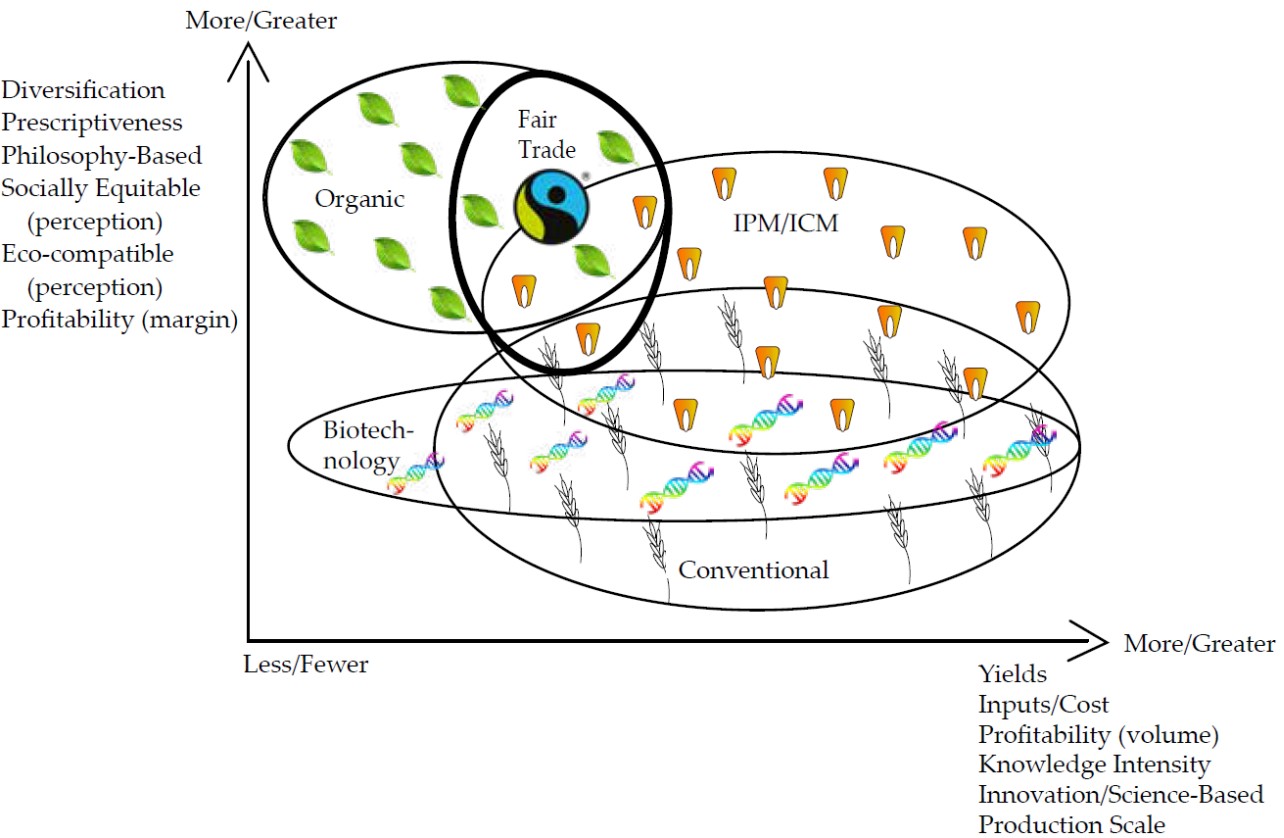

**Figure 1.** Characteristics of Competing Production Typologies. Source: authors' composition based on the analyzed literature.

### 3.2.3. Ecological Economics

Amid the backdrop of carrying capacity, environmental limits, and sustainability, the obstinate question of whether farms are "a part" or "apart" from nature persists. Bliss and Egler (2020) contend that markets are artificial constructs and inappropriate in most relational contexts to ecological economists, nor do they serve justice, sustainability, efficiency, or value pluralism.

Despite this, the tendency among agricultural economists is to quantify and monetize. Competing production schemas are often assessed on three factors: environmental impacts, energetics, and economics. Invariably, the latter two are embodied in an economic assessment.

In addition to typical economic indicators (production costs including inputs, marketing, and wholesale/retail consumer margins) the intersection of economics and the environment deserves careful consideration. Specifically, the monetization of ecosystem services afforded by the various production systems: ecological economics.

Naranjo et al. (2019) provide an exhaustive review of the economic valuation of arthropod biological control in both natural and (agro)ecosystems. Broad estimates suggest that biological control services are valued at USD 619/ha across multiple biomes in natural settings, while the estimated value is USD 36/ha in croplands. Natural biological control of native US crop pests is estimated at USD 5.95 billion, with some evidence suggesting this is grossly undervalued. Invariably, a link exists between a crop's market value and the resultant worth of biological control, allowing farmers to gauge the highest potential returns on investment in an IPM framework.

Ecosystems services in a broader context are often ill-defined and may outright defy quantification. For example, system effects on pollinators and other beneficial insects, and the short/long-term value of conserving those actors in-field. Despite the initial time and cost outlays, there's intrinsic cost savings (reduced pesticide costs) and net gains by

providing refugia. For instance, borders planted in wildflowers to conserve beneficials in lieu of herbicide use.

Meadows in particular are a popular target for conservation. They are less actively managed agroecosystems, with a greater degree of permanence and stability than traditional fields. As such, they enjoy greater biodiversity and are targets for conservation programs. They also represent a forum to examine the comparative effects of production practices on biodiversity. Seeking to delineate these differences, Gerling et al. (2019) used an ecological-economic modelling procedure to analyze management approaches on endangered bird and butterfly species in Germany. The impact of conventional and organic farming alone was minor, because land use timing of both systems aligns. However, organic farmers faced lower opportunity costs when implementing agri-environment scheme (AES) measures. Organic farmers were also offered lower payments, which can disincentivize their participation in AES; in turn suppressing conservation impacts.

While ecosystem services are doubtless important, negative externalities can manifest if there's an overreliance on these amenities. For example, while organic production systems may be considered more eco-friendly, this typically only applies to the micro, farm level. When viewed in aggregate, the organic yield penalty can represent a perverse financial incentive. Specifically, one that necessitates the compensatory conversion of wild habitat to agriculture to farmland.

Indeed, system "reconciliation" can be infinitely confounding. Conventional agriculture has embraced the use of no-till systems using herbicide tolerant, biotech varieties (e.g., Roundup Ready, LibertyLink, etc.). Though the use of spray-intensive systems is frequently criticized as reducing microbial diversity (which can adversely impact "free" nutrient cycling), it comes with a suite of benefits, including the reduction of erosion and sequestration of carbon. The latter could be incentivized by governments as part of a Carbon Scheme to pay farmers as a best practice, thus providing an alternative and predictable revenue stream to combat the vagaries of production.

Similarly, land scarcity is another factor to consider as prime farmland is converted to human spaces. Land represents convertible value to the farmer if s/he were to site green energy—wind turbines and/or solar panels, etc., especially in concert with (or as a successor to) a silvopastoral system.

### 3.2.4. Socioeconomic Contributors to the Regulation of Pest Management

Waterfield and Zilberman (2012) outline some of the externalities of pest management, including environmental effects, public/occupational health, and even productivity effects on neighboring farms. The magnitude of these effects differs widely across pest management technologies and the situations to which they are applied. An optimal pest management calculus is a careful balancing act. Ideally, it strives to balance the quantifiable benefits of yield, while considering risk reduction against external costs, particularly nonpecuniary characteristics that impact farmers' decisions and welfare. Such analysis should be the basis of government regulation.

Travisi et al. (2006) presented a critical overview of pesticide risk valuation that provides disaggregate willingness-to-pay estimates (WTPs) of pesticide risk reduction. They used recent multidimensional classification methods, including coined decision tree analysis, as tools in a comparative approach to account for differences in empirical research findings. The analysis showed that the order of magnitude of WTPs is related to both the valuation technique and to the data available from biomedical and ecotoxicological literature. It also shows that WTP estimates of pesticide risks cannot be averaged over several empirical studies. The order of magnitude of a WTP estimate is related to the specific type of risk, the nature of the risk scenario considered, as well to lay people's subjective perception of said risk.

According to Sexton et al. (2007) economics can be used to inform both private and public decision-makers about trade-offs inherent in pesticide use and other pest management strategies. The betterment of social welfare has generally been the impetus. Deter-

minations focus on balancing the benefits against (reduced crop injury) with total costs. These include those borne by the farmer (price and application cost) and those imposed on society (risks to human health and degraded environmental quality). Such analyses consider a host of issues, including externalities and uncertainty. Despite this, economists have developed valuable and robust methodological tools to inform decision-makers.

Solomon (2015) noted that a multitude of factors must be considered by agencies tasked with pesticide approval and oversight. Several socioeconomic analysis techniques can be used to quantify the full spectrum of issues and help improve management, including the adequate consideration of alternatives. The most popular and commonly used techniques are cost–benefit analysis and cost-effectiveness analysis. Further, another family of methods, known as Rapid Rural Appraisal and Participatory Rural Appraisal, can be more appropriate, faster, and have lower cost to use in developing countries. Finally, there is qualitative decision making under uncertainty, such as the use of the Precautionary Principle. Ideally, all of the analytical techniques will need complete and reliable socioeconomic data, though in reality, data are often incomplete and fraught with uncertainty. In these cases, the application of the Precautionary Principle decision rule may have strong justification.

However, the Precautionary Principle, taken to its logical extreme, can lead to a regulatory body opting to dismiss the benefits of a pesticide while unduly magnifying risks. This is further complicated when considered within an IPM context, where fewer applications and potential exposures are likely.

## 4. Discussion and Conclusions

Agricultural production systems can significantly differ in motivations and execution. Occasionally they are spatially separated; more often they exist side-by-side. Most of the literature compared and contrasted organic and conventional production. There is a clear consensus that organic production normally results in much lower input costs, higher labor need, lower yields, and higher prices compared to a conventional system. Diversification helps to reduce income risks, as well as to balance input purchases and product marketing. Naturally, these differences vary greatly from product to product or country to country. Agronomic crops normally perform better than fruits, vegetables, and livestock. Country-level comparisons demonstrated that organic farms usually have better financial performance. In addition, organic production provides environmental benefits at the micro level that is often challenging to monetize. With these enticements in mind, the transition from conventional to organic production is a unique issue as farmers face all the disadvantages (higher labor cost and lower yield) without any price premium in the interim. It should be noted that there are diversified farming systems that, while inspired by a given system, do not cleanly align. For example, biodynamic farms even surpass the strict production rules of organic farming regarding the use of herbal and mineral additives.

IPM is a system agnostic potentiator. It is a broadening approach that employs decision-making informed by scouting and economic thresholds. Injury and damage abatement tools abound, and are often used in harmonious concert. It can represent considerable cost savings and ecological enhancements. However, it is also labor and knowledge intensive, a potentially daunting constraint for farmers with limited technical support (e.g., Extension services).

Biotechnology is a relatively new approach. In areas of widespread adoption, it has often paralleled the industrial model, yet garners environmental benefits. Based on the reviewed articles, GM typically provides lower production cost, higher yield, and more efficient production. However, there are concerns about horizontal gene transfer and resistance, potential allergenicity, and dependency on seed providers. The economic benefits are mainly at the farm level, stemming from lower pesticide use, (therefore) reduced fuel and labor need, and higher yields. It should be noted that GM crop technologies are scale neutral, thus smaller and lower income farmers may achieve higher gross margin. Additionally, the use of "no-till" and "reduced-till" farming practices that leverage envi-

ronmentally friendly herbicidal chemistries facilitate considerable gains in the arenas of soil erosion and carbon sequestration.

The main issues confronting biotechnology are (organic) coexistence and labeling. The maintenance of seed purity is especially important for organic farmers as contamination sabotages their price premium, though the legal concept of adventitious presence allows for a minor degree of adulteration without penalty (below a set threshold). In addition, labeling is often viewed as a consumer's right to know, especially in those countries more skeptical of GM-based foods.

Table 1 provides a summary of the major characteristics of these different crop production systems.

**Table 1.** Major Characteristics of Crop Production Systems.

| Characteristics | Conventional | Conventional Plus Biotechnology | Organic | IPM |
|---|---|---|---|---|
| Yield | Normal | Up to 15% higher | At least 10% lower | Comparable to Conventional |
| Pesticide Cost | Normal | Lower | Much lower | Much lower |
| Fertilizer Cost | Normal | Normal | Much lower | Normal |
| Labor Cost | Normal | Lower | Higher (+15%) | Higher |
| Product Variety | Specialization | Specialization | Diversification | Diversification |
| Product Price | Normal | Normal | High price premium | Normal |
| Gross Margin | Normal | Much higher | Generally higher | Higher |
| Pricing/Business Model | Volume | Volume | Margin | Margin |
| Environmental Benefits | Normal | Higher | Much higher (at the micro level) | Much higher |

Source: authors' composition based on the analyzed literature.

The major limitation of this study was the selection method. Other authoritative sources can be used, e.g., Scopus, ISI Web of Science, JSTOR, and ProQuest. However, the value of additional databases is nominal, as the majority of articles have an overlapping occurrence. Keyword searches can also be structured to further leverage and diversify content for inclusion.

Moreover, some referenced studies attributed disparities in organic/conventional profitability using different crops, a figurative apples-to-oranges comparison. Though this may or may not be representative of prevailing trends, single-crop standardization would be more prudent and reflective of market-driven differences.

Overall, competing production schemas are generally assessed on three factors: environmental impacts, energetics, and economics. Invariably, the latter two are embodied in an economic assessment. Although this investigation specifically focused on the comparative economics of "competing" agricultural systems, attempts to monetize energetics (briefly alluded to) and environmental impacts provide a valuable corollary to the discussion.

With the exception of organics, farmers are not singularly empowered nor restricted to use a given technological innovation or practice. Yet, the returns associated with organics has genuine appeal for entrenched farmers and newcomers. Despite that, production systems are largely artificial constructs, and considerable cross functionality exists between them. Thus, system distinctions can quickly become muddled. Indeed, it's largely an artifact of marketing.

A frequent mistaken notion is that specific practices are system exclusive. The use of IPM and/or biotechnology in a system otherwise identified as conventional is a case in point. Another example is the use of cover cropping, rotation, and biocontrols in a conventional framework. Or a conventional system that practices diversified cropping in discrete, small blocks (e.g., fractional monoculture). Alternatively, a conventional/IPM system may identify as "biointensive" and use biorational (naturally derived pesticides), but also utilize synthetic legacy chemistries that have largely been phased out—on a calendar basis no less—because no useful substitute exists, and the IPM threshold for the pest (or pathogen responsible for disease) is essentially zero. Moreover, USDA organic systems do permit the use of select synthetic materials, as per the Organic Materials Review Institute (OMRI) standards.

The reality is that production philosophies are rarely as rigid and orthodox as inferred. Most are a composite, guided by the respective farmer's worldview, available resources, market/clientele demand, and financial expediency. This confounds efforts to fully capture the economic costs and ecological valuations of competing typologies.

The next 30 years are a critical juncture in human history. Merely keeping pace with population growth will necessitate significant gains in agricultural output per unit area. To satisfy this projected growth—without substantial extensification of farmland—agricultural yields must commensurately increase. As such, sustainable pathways for bolstering productivity, farmer income, and food security/safety are paramount. No single production typology is the panacea to this challenge, especially in light of farm-specific circumstances. The empirical literature suggests that a composite system can embody ecological reverence, social justice, and economic fundamentals—the requisite elements in an often-bewildering search for reconciliation.

Indeed, farmers should focus on an a la carte approach when possible, with site-specific optimization of practices and marketing. In addition to aforementioned approaches, this can include the planting of perennialized crops, independent labeling programs, the formation of cooperatives, buy local initiatives, as well as community supported agriculture and U-Pick, where applicable. All of these exist outside of the orderly confines of an organic/conventional dichotomy.

**Author Contributions:** Conceptualization, T.C.D. and T.M.; methodology, T.C.D. and T.M.; investigation, results and conclusions, T.C.D. and T.M.; writing—original draft preparation, T.C.D. and T.M.; writing—review and editing, T.C.D. and T.M.; visualization, T.C.D. and T.M.; supervision, T.C.D.; funding acquisition, T.M. All authors have read and agreed to the published version of the manuscript.

**Funding:** This research received no external funding.

**Institutional Review Board Statement:** Not applicable.

**Informed Consent Statement:** Not applicable.

**Data Availability Statement:** Not applicable.

**Acknowledgments:** The authors wish to thank the Economies Editorial Office this publication opportunity.

**Conflicts of Interest:** The authors declare no conflict of interest.

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
