# Peer review of "Comparative Economics of Conventional, Organic, and Alternative Agricultural Production Systems"

_economies, doi:10.3390/economies9020064_

Round 1

Reviewer 1 Report

The proposed literature review is very interesting and easy to read, with a fairly fluid and well-organised structure. I suggest the authors to specify the methodological criterion for the selection of the 82 papers before and 63 papers after. 
Immediately after the methodology the authors start the description of the results according to a good organizational structure, however I suggest the authors to insert a table, also supplementary material, in which are described all the papers analyzed according to the parameters that then have been analyzed in the paragraph of the results and to enrich all the subsections with graphs or tables that allow to have an overview of the in-depth topic.

Author Response

Dear Reviewer 1,

Many thanks for your detailed comments and suggestions which helped us to revise our manuscript. We have responded to your comments in the following way:

The proposed literature review is very interesting and easy to read, with a fairly fluid and well-organised structure. I suggest the authors to specify the methodological criterion for the selection of the 82 papers before and 63 papers after.

Added rationale in appropriate section.

Immediately after the methodology the authors start the description of the results according to a good organizational structure, however I suggest the authors to insert a table, also supplementary material, in which are described all the papers analyzed according to the parameters that then have been analyzed in the paragraph of the results and to enrich all the subsections with graphs or tables that allow to have an overview of the in-depth topic.

We constructed and added a Venn Diagram for inclusion (visually compares and contrasts the referenced systems).

Overall, thanks again for all your comments which helped us to revise and significantly improve the manuscript. We hope now you think the paper is ready for publication.

Reviewer 2 Report

Overall, you make a good point on each system. In the end your conclusions are dependent on the strength of each of the studies you include. Although a study might get published it may have inherent problems. Several of the papers sited appear to make conclusions on differences that may be more a result of different crops rather than comparing conventional to organics. Income differences could be due to one being a higher value crop (alfalfa commands a higher price than say wheat as a forage). Comparing organic alfalfa production to conventional wheat will almost always end up with alfalfa winning.

Editorial suggestions below:

Line 149 - change "organizing" to "organizational"

Line 179 - change "instance" to "instances"

Line 379 - "prescriptions" should be changed to "regulations"

Line 500 - remove "build"

Author Response

Dear Reviewer 2,

Many thanks for your detailed comments and suggestions which helped us to revise our manuscript. We have responded to your comments in the following way:

Overall, you make a good point on each system. In the end your conclusions are dependent on the strength of each of the studies you include. Although a study might get published it may have inherent problems. Several of the papers sited appear to make conclusions on differences that may be more a result of different crops rather than comparing conventional to organics. Income differences could be due to one being a higher value crop (alfalfa commands a higher price than say wheat as a forage). Comparing organic alfalfa production to conventional wheat will almost always end up with alfalfa winning.

We added short paragraph to address disparate comparisons under Discussion and conclusions. However, our keywords-based search limited the number of potential articles.

Editorial suggestions below:

Line 149 - change "organizing" to "organizational"

Line 179 - change "instance" to "instances"

Line 379 - "prescriptions" should be changed to "regulations"

Line 500 - remove "build"

We corrected all of them.

Overall, thanks again for all your comments which helped us to revise and significantly improve the manuscript. We hope now you think the paper is ready for publication.

Reviewer 3 Report

Thank you very much for the opportunity to read "Comparative Economics of Conventional, Organic, and Alternative Agricultural Production Systems."

I note that the structuring of the article is adequate, including Introduction, Methodology, Comparative economics of agricultural production systems, Discussion and conclusions.

Overall, the paper would have had greater value if:

- use a mathematical model;

- is based on articles published in prestigious ISI Web of Sci journals, published more recently (2019, 2020, 2021).

The paper is acceptable as a theoretical study, provided that the editors of the journal accept such papers.

In this situation, the documentation and bibliography should be extended on the basis of articles published in prestigious ISI Web of Sci journals, published more recently and possibly supervised by a native English teacher.

Author Response

Dear Reviewer 3,

Many thanks for your detailed comments and suggestions which helped us to revise our manuscript. We have responded to your comments in the following way:

Thank you very much for the opportunity to read "Comparative Economics of Conventional, Organic, and Alternative Agricultural Production Systems." I note that the structuring of the article is adequate, including Introduction, Methodology, Comparative economics of agricultural production systems, Discussion and conclusions.

Overall, the paper would have had greater value if:

- use a mathematical model;

- is based on articles published in prestigious ISI Web of Sci journals, published more recently (2019, 2020, 2021).

The paper is acceptable as a theoretical study, provided that the editors of the journal accept such papers. In this situation, the documentation and bibliography should be extended on the basis of articles published in prestigious ISI Web of Sci journals, published more recently and possibly supervised by a native English teacher.

Mathematical models are outside of our purview. We have added one more article. However, we ran keywords-based search that provided a limited number of articles. We also rationalized our use of ScienceDirect and Google Scholar (and not others) in the methodology section, as well as the potential for ”overlapping occurrence” using other databases in the discussion/conclusion section.

On the whole, thanks again for all your comments which helped us to revise and significantly improve the manuscript. We hope now you think the paper is ready for publication.